# Three-Dimensional Printing of Large Objects with High Resolution by Dynamic Projection Scanning Lithography

**DOI:** 10.3390/mi14091700

**Published:** 2023-08-30

**Authors:** Chunbo Lin, Wenbin Xu, Bochao Liu, He Wang, Haiping Xing, Qiang Sun, Jia Xu

**Affiliations:** 1Research and Development Center of Precision Instruments and Equipment, Changchun Institute of Optics, Fine Mechanics and Physics, Chinese Academy of Sciences, Changchun 130033, China; linchunbo20@mails.ucas.edu.cn (C.L.); xuwenbin@ciomp.ac.cn (W.X.); liubochao@ciomp.ac.cn (B.L.); whciomp@163.com (H.W.); sunq@ciomp.ac.cn (Q.S.); 2University of Chinese Academy of Sciences, Beijing 100049, China; 3State Key Laboratory of Polymer Physics and Chemistry, Changchun Institute of Applied Chemistry, Chinese Academy of Sciences, Changchun 130022, China; hpxing@ciac.ac.cn

**Keywords:** 3D printing, additive manufacturing, DLP, digital super resolution

## Abstract

Due to the development of printing materials, light-cured 3D printing is playing an increasingly important role in industrial and consumer markets for prototype manufacturing and conceptual design due to its advantages in high-precision and high-surface finish. Despite its widespread use, it is still difficult to achieve the 3D printing requirements of large volume, high resolution, and high speed. Currently, traditional light-cured 3D printing technologies based on stereolithography, such as regular DLP and SLA, can no longer meet the requirements of the processing size and processing rate. This paper introduces a dynamic projection of 3D printing technology utilizing a digital micro-mirror device (DMD). By projecting the ultraviolet light pattern in the form of “animation”, the printing resin is continuously cured in the exposure process to form the required three-dimensional structure. To print large-size objects, the three-dimensional model is sliced into high-resolution sectional images, and each layer of the sectional image is further divided into sub-regional images. These images are dynamically exposed to the light-curing material and are synchronized with the scanning motion of the projection lens to form a static exposure pattern in the construction area. Combined with the digital super-resolution, this system can achieve the layering and fine printing of large-size objects up to 400 × 400 × 200 mm, with a minimum feature size of 45 μm. This technology can achieve large-size, high-precision structural printing in industrial fields such as automobiles and aviation, promoting structural design, performance verification, product pre-production, and final part processing. Its printing speed and material bending characteristics are superior to existing DLP light-curing 3D printing methods.

## 1. Introduction

Three-dimensional printing technology, also known as additive manufacturing technology, has been a rapidly developing green high-end manufacturing technology in recent years and is one of the three key technologies for realizing digital manufacturing along with artificial intelligence and intelligent robots [1,2]. Unlike traditional subtractive manufacturing processes such as cutting processes like turning, planning, grinding, and drilling in CNC machining, 3D printing technology does not obtain the desired shape by “subtracting” from materials. Instead, it is based on the three-dimensional digital model of the target structure and uses a layer-by-layer manufacturing and accumulation method to “add” materials to achieve three-dimensional processing [3]. It has technological advantages such as digitalization, intelligence, and customization [4,5,6,7]. It realizes a continuous and adjustable printing structure from micro to macro and a controllable printing process [8]. This manufacturing method can be widely used in aerospace, industrial equipment, bio-medical, consumer electronics, and automotive industries [9,10]. It achieves controllable microscopic material organization, adjustable macroscopic structural performance, and a controllable manufacturing process, and has technological advantages such as low cost, high integrity, and high structural accuracy [11,12,13]. Meanwhile, the light-cured 3D printing technology has the advantages of high manufacturing accuracy, fast manufacturing speeds, diverse material selection, and high design freedom, and can not only manufacture high-precision and complex parts but also has applications that other 3D printing technologies cannot achieve, including high-resolution micro-manufacturing, high-performance medical equipment and artificial organs, rapid prototyping, and the manufacturing of optical and electronic devices [14,15,16,17].

Currently, 3D printing using photo-curing can be divided into three categories based on the exposure process: stereolithography (SLA) 3D printing technology, liquid crystal display (LCD) 3D printing technology, and digital light processing (DLP) 3D printing technology. SLA 3D printing is the first-generation light-cured 3D printing technology [18,19,20,21]. Its basic principle is to use ultraviolet lasers or LEDs as the light source and control the light spot to scan and expose the surface of the liquid printing material through a mirror system, completing one layer of solidification. The build platform is then lowered by a layer of thickness, and the process is repeated, layer by layer until the solid object is completed. The main advantage of this method is its high printing resolution; however, the scanning method greatly reduces the printing speed, and the use of a mirror system limits the scanning range and reduces the consistency of edge and center resolution in the printing area. The basic principle of LCD 3D printing is to generate graphic information through a computer program, and the liquid crystal screen forms selectively transparent areas based on graphic information [22,23,24]. Ultraviolet light passes through these areas and shines on the liquid printing material from the bottom up, solidifying the surface. After each layer is printed, the build platform moves up a layer in thickness, and this process is repeated layer by layer until the solid structure is completed. LCD 3D printing uses the liquid crystal screen as a graphic generator to perform surface exposure on the printing material, greatly increasing the printing speed compared to SLA’s point-by-point scanning exposure. However, since the resolution of the printing structure depends entirely on the pixel size of the LCD screen, it cannot achieve high-precision 3D model printing. DLP 3D printing is second-generation light-cured 3D printing technology, which mainly uses a projection system to solidify the printing material layer by layer to quickly form a three-dimensional printing structure [25,26,27]. First, the model is sliced using slicing software to obtain a two-dimensional cross-sectional image. Each layer of the image is projected onto the surface of the printing material to form a light pattern, which solidifies the surface. After each layer is printed, the platform moves down a layer, and the system continues to project the next layer of light patterns for printing until the process is complete. Since a digital micro-mirror device is used as the graphic generator, this method not only has high molding accuracy but also a fast printing speed, making it possible to quickly print models with complex structures and precise details. However, due to the utilization of projection exposure for layer-by-layer printing, the printing area is solely determined via the system’s projection range, which restricts the achievement of high-precision printing for large-volume models.

Dynamic projection scanning (DPS) 3D printing technology uses a digital micro-mirror device as the graphic generator. The graphic information is converted into electrical signals to control the surface micro-mirrors of the DMD through a computer program. Ultraviolet light emitted by the light source is reflected by the DMD surface micro-mirror array to form the required dynamic light patterns. The frame rate of the graphic on the DMD is synchronized with the scanning speed of the exposure objective lens. By employing this method, the exposure objective lens scans and dynamically exposes the printing material surface, allowing for the completion of one printing layer at a time. This process forms the basis of dynamic projection scanning 3D printing technology, which enables the high-precision printing of large-volume models.

## 2. System and Methods

As shown in Figure 1, it is a dynamic projection scanning 3D printing exposure system, which mainly consists of an illumination system, a photolithography engine, a scanning motion control system, and a sample stage. The illumination system includes a 405 nm ultraviolet laser light source with a continuously adjustable power of 100 mW to 1000 mW and a beam shaper. The photolithography engine mainly includes a DMD (digital micro-mirror device) and a projection lens, with a DMD micro-reflection mirror pixel resolution of 1920 × 1080 and a single pixel size of 10.8 microns. The projection exposure lens with a magnification of ×1 has a numerical aperture of 0.086. The scanning motion control system has a motion accuracy of 1 μm and positioning accuracy of 0.5 μm and is used to drive and control the photolithography engine to achieve scanning exposure in the two-dimensional direction.

The ultraviolet light emitted by the light source is shaped and collimated by the illumination system before entering the DMD surface. By modulating the digital micro-mirror array (DMD), the reflected light is selectively directed into the projection lens, creating the desired dynamic light pattern that is projected onto the surface of the sample stage. The exposure system is driven and controlled in the X-Y direction using a linear motor, and its movement speed is synchronized with the frame rate of the dynamic pattern displayed on the DMD. This high-speed scanning exposure technique enables the acquisition of a large-area printing pattern.

Due to the continuous movement of the lithography engine during the exposure process, the pattern on the DMD must roll continuously along the scanning direction and be synchronized with the scanning speed of the lithography engine to form the required static pattern on the substrate surface. As shown in Figure 2, under continuous illumination, when the lithography engine moved at a one-pixel distance, the pattern on the DMD rolled one row. According to the given pattern, the micro-mirrors on the DMD surface are dynamically flipped in a one-by-one manner. The reflected light passes through the projection lens to form the required pattern structure on the substrate. However, the use of square micro-mirrors on the DMD greatly reduces the lithography resolution of this traditional scanning exposure mode, resulting in serious “staircase” phenomena at the edges of the exposed patterns. When the feature size of the pattern to be processed is close to the pixel size, the resulting pattern structure experiences significant distortion. To reduce the influence of the edge roughness caused by pixel quantization errors in traditional DMD scanning lithography, we used digital super-resolution imaging to achieve sub-pixel level rolling exposure through multiple pixel stacking exposures, as shown in Figure 3.

The digital super-resolution imaging based on DMD oblique scanning utilizes the angle between the DMD image movement step and the scanning motion direction of the lithography engine, combined with the DMD mirror array loading and control algorithm, to obtain oblique scanning exposure with a digital resolution of R, and prepares the sub-pixel feature-size pattern structures on the sample surface. Utilizing GerbMagic software V3.7, the original vector image was subjected to high-resolution rasterization within specified regions. Subsequently, image rotation and pixel merging were applied to the rasterized pixel matrix in order to generate a projection image in the coordinate system for display on DMD. The basic principle is shown in Figure 4, where Y is the scanning and rolling exposure direction of the lithography engine, X is the lateral movement direction of the lithography engine, P is the center distance between adjacent mirrors on the DMD, and θ is the oblique angle of the DMD. In actual exposure, DMD updates the exposure pattern with every relative displacement of P along the scanning direction to achieve the scanning effect with the resolution:R = P∙sin(θ)(1)

To ensure the multiple relationships between image segmentation in oblique scanning, the oblique angle of the DMD needs to satisfy:Tanθ = 1/N,(2)
where the tilting factor N is a positive integer. The larger the N, the smaller the oblique angle θ and the smaller the digital resolution obtained, which results in the higher precision of the scanning exposure pattern.

Traditional DMD scanning involves arranging the micro-mirror array parallel to the exposure scanning direction, which means that the gaps between the micro-mirrors can only rely on the dispersion of light in the printing material for exposure, leading to inconsistent exposure between the directly illuminated areas and the gaps. By using oblique scanning, the gaps between the micro-mirrors and the scanning direction can be angled, and the gaps between the pixels are covered by the tilted micro-mirrors during dynamic scanning, avoiding the influence of micro-mirror gaps on the exposure measurement of the printing material. At the same time, using a large magnification and depth for the focus lens in tilted scanning can further unleash the printing capabilities of traditional straight scanning with a small magnification and depth of focus lens, releasing the performance of the optical engine. As shown in Figure 5, we conducted 3D printing experiments on the surface of the resin material using traditional DMD scanning exposure and super-resolution digital exposure. The experimental results indicate that 3D printing based on traditional DMD scanning exposure resulted in a surface covered with groove structures at the pixel-level periodicity, significantly affecting the printing accuracy and structural integrity. Conversely, the digital super-resolution 3D printing based on DMD oblique scanning produced a smooth surface with a printing resolution that far exceeded the resolution of the resin material, making it suitable for fabricating high-precision models.

During the scanning and exposure process, after each column of scanning and exposure was complete, the lithography engine moved horizontally along the X-axis at a certain distance before performing the next column of scanning and exposure. The projection shape of the DMD micro-mirror array was rectangular, and in tilted scanning, the rectangular projection of the DMD was divided into two triangular regions and one parallelogram region by the scanning direction. As shown in Figure 6, the scanning positions of the triangular regions between adjacent scanning and exposure columns overlapped. The exposure energy of each pixel in the triangular region came from the overlap of the adjacent triangular regions, and its accumulated energy was equivalent to the pixels in the parallelogram region. This scanning and exposure method with region overlap effectively eliminated the edge effect of the lens, and the position of the graphics stitching changed from the traditional narrow gap to a stitching area with a certain width, where the exposure intensity was more uniform, largely eliminating the mechanical performance impact of stitching on the printed structure.

## 3. Results and Discussion

We utilized the UV-photosensitive resin produced by Creality as the printing material, with a photosensitive wavelength range of 355–410 nm, a forming density of 1.25 g/cm^3^, and a volume shrinkage rate of 3.72%. With the use of this technology, models were successfully manufactured with a diverse range of dimensions. As illustrated in Figure 7a, a Chinese dragon measuring 400 mm × 200 mm in the XY plane was produced. This model size showcases the remarkable large-scale production capabilities that are typically unattainable through conventional DLP-based 3D printers. Figure 7b exhibits the printing of a typical 3D model, the Eiffel Tower, which features complicated details, including micro-scale structures. Notably, the vertical rails in Figure 7c possess a width of 45 μm. The successful printing of the Eiffel Tower model demonstrates this technology’s capability to produce complex structures with robust inter-layer bonding. Additionally, Figure 7d–g showcases the printing of a collection of characteristic models within a short time frame of 3 h.

To evaluate the feature size that could be fabricated by this technology, objects with micro-scale features were printed and are presented in Figure 8. As Figure 8a shows, vertical-wall structures with different gap distances were printed, achieving a minimum gap of 80 μm with well-defined boundaries. Additionally, Figure 8b illustrates the capability of this technique to create micro-channel structures; a micro-channel sample with a hexagon cross-section was used as an example, and the microscope image of the cross-section confirmed the information of a uniform micro-channel with a wall thickness of 100 μm.

Figure 9 presents the comparisons of the breaking bending load and maximum bending load among the different printing methods. In the test, all samples were printed using the same resin material. Both traditional scanning printing and dynamic scanning printing employed the same exposure process: an exposure energy of 950 mW/cm^2^ and a printing scanning speed of 16 mm/s. On the other hand, step-by-step printing employed a distinct exposure process: an exposure energy of 950 mW/cm^2^, an exposure time of 1.25 s, and a step distance of 11.6 mm. Moreover, the printed samples for these aforementioned three printing methods had identical dimensions of 20 mm × 10 mm with a thickness of 2 mm. Utilizing an INSTRON-5869 material testing machine under room temperature conditions, three-point bending tests were individually conducted on samples obtained from these three printing methods. It can be observed that, compared to the other two printing methods, projection exposure 3D printing yielded a significantly improved mechanical strength in the printed results. These results served as compelling evidence for the superior capabilities of our technology, as it not only enabled the production of larger parts but also ensured their enhanced strength.

## 4. Conclusions

This paper introduces a dynamic projection scanning lithography technology that enables the manufacturing of large-scale parts with high-resolution features and high-speed 3D printing. Using this technology, a series of prototypes were developed, demonstrating the capability of this technology to achieve a minimum feature size of 45 μm and a build volume of 400 × 400 × 200 mm. This scalable 3D printing method allows for the production of larger customized build volumes while retaining micro-scale features and a smooth finish. Compared with traditional DLP-based scanning 3D printing technology, we employed the dynamic projection exposure technique based on a digital super-resolution to decrease the impact of the “staircase” effect on the printing results. Based on experimental testing, it was established that the printed models exhibited significantly improved mechanical strength compared to conventional scanning 3D printing and stepwise printing methods. As a result, they can be widely utilized in various sectors, including aerospace, industrial equipment, consumer electronics, and digital healthcare.

## Figures and Tables

**Figure 1 micromachines-14-01700-f001:**
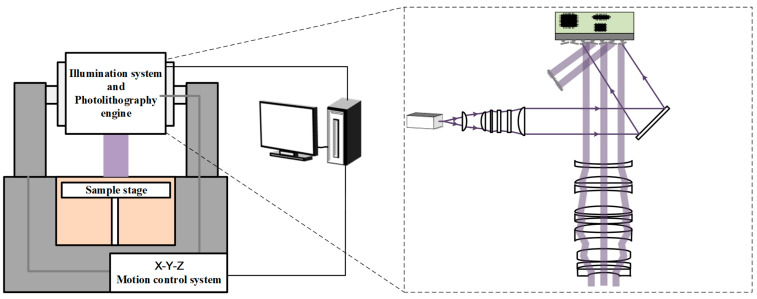
Schematic diagram of the dynamic projection scanning 3D printing exposure system.

**Figure 2 micromachines-14-01700-f002:**
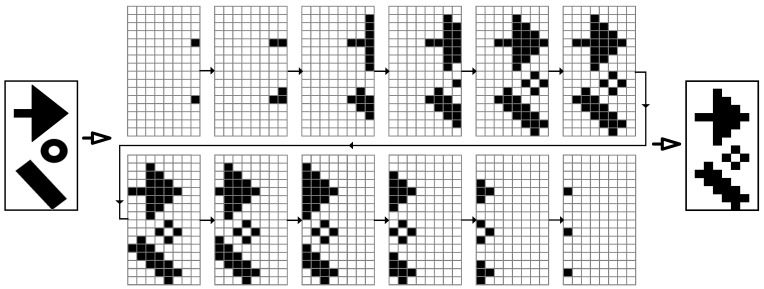
The illustration of frame divisions.

**Figure 3 micromachines-14-01700-f003:**
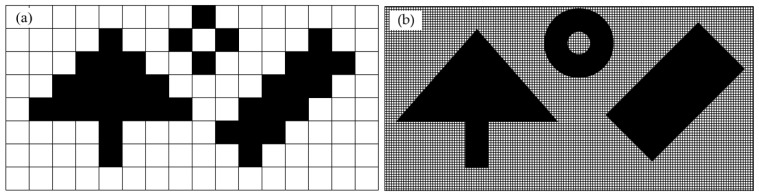
Comparative illustration of traditional scanning lithography and our dynamic scanning lithography: (**a**) Pixel-level resolution traditional DMD lithography; (**b**) Sub-pixel-level resolution dynamic scanning lithography.

**Figure 4 micromachines-14-01700-f004:**
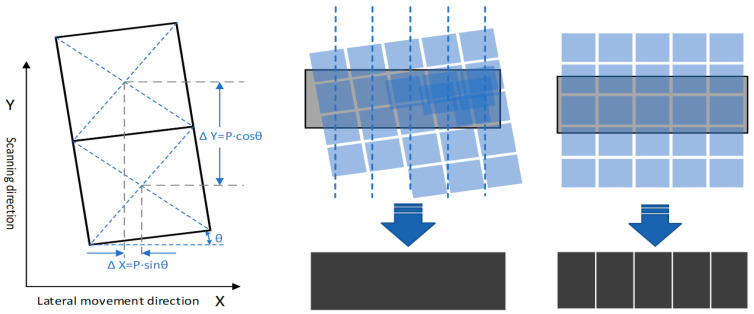
Schematic diagram of digital super-resolution imaging based on DMD oblique scanning vs. pixel-level resolution based on traditional DMD scanning.

**Figure 5 micromachines-14-01700-f005:**
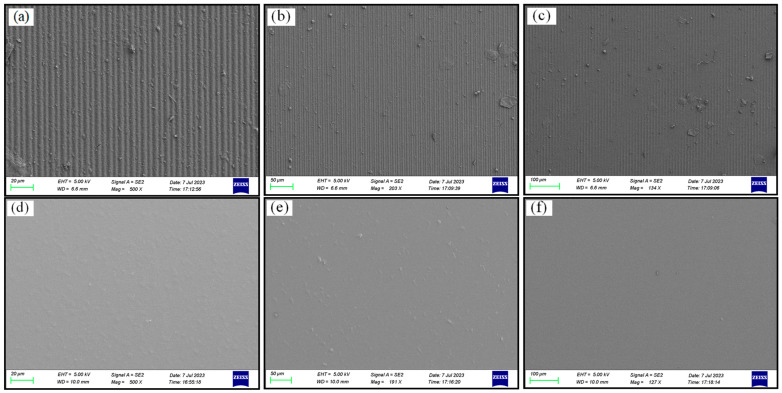
SEM micrographs of 3D-printing results: (**a**–**c**) Groove structures obtained from traditional DMD scanning printing, magnified by 500×, 190×, and 135×; (**d**–**f**) Flat structures based on DMD oblique scanning printing, magnified by 500×, 190×, and 135×.

**Figure 6 micromachines-14-01700-f006:**
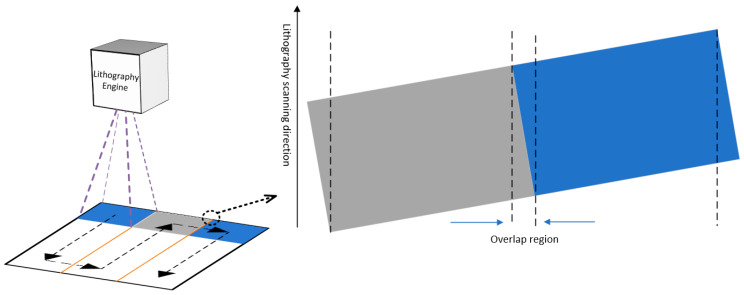
Illustration of scanning pathway based on dynamic scanning exposure.

**Figure 7 micromachines-14-01700-f007:**
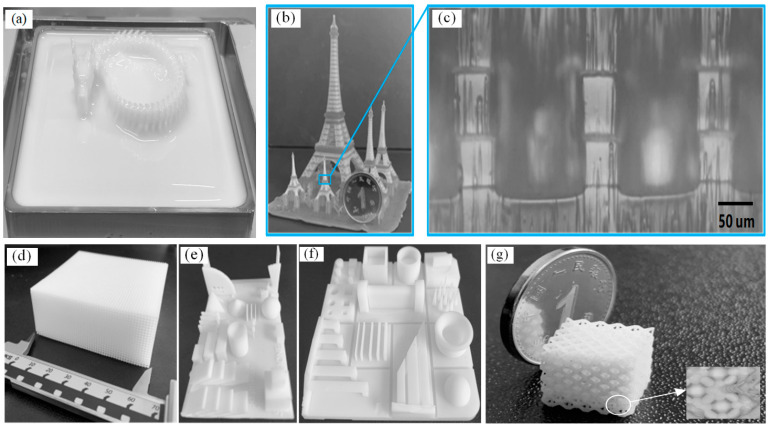
Printed samples: (**a**) a Chinese dragon; (**b**,**c**) The Eiffel Tower and its vertical rails; (**d**) A hollow structure; (**e**–**g**) as characteristic models.

**Figure 8 micromachines-14-01700-f008:**
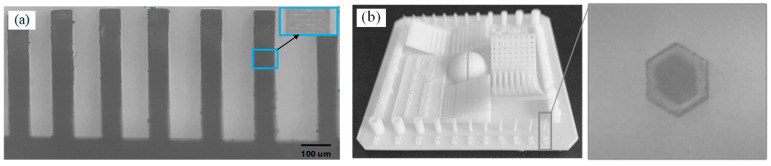
(**a**) 3D printed vertical walls with a minimum gap of 80 μm; (**b**) Micro-channel with a wall thickness of 100 μm.

**Figure 9 micromachines-14-01700-f009:**
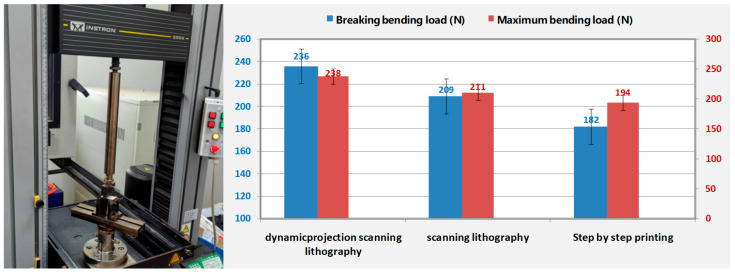
Comparison of maximum bending load and fracture bending load under the same structural conditions for three different printing technologies printing the same material.

## Data Availability

Not applicable.

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
