# Peer review of "Three-Dimensional Printing of Large Objects with High Resolution by Dynamic Projection Scanning Lithography"

_micromachines, 2023, doi:10.3390/mi14091700_

Round 1

Reviewer 1 Report

The paper introduces dynamic projection scanning lithography as a method for high-resolution 3D printing of large objects, and while the results show potential, there is room for improvement in terms of adequately highlighting the novelty of the proposed work compared to the current state-of-the-art in dynamic projection scanning lithography.

1.       It would be beneficial to include references and comparisons with updated reports in the field to establish the paper's benchmark. Consider incorporating a comparison table to emphasise the superior features of the proposed work. Some references can be compared, such as

•         https://www.nature.com/articles/srep09875

•         https://iopscience.iop.org/article/10.1088/2631-7990/ab8d9a

•         https://www.sciencedirect.com/science/article/pii/S2214860419301666

2.       Additionally, providing extensive information on how digital super-resolution imaging was employed in this study, including details about any relevant software used, will enhance clarity.

3.       To further enhance the paper's value, it would be valuable to include a section discussing the cost of the entire process, as this aspect is important for readers' understanding.

4.       For Figures 3 and 4, clear differentiation should be made between the outcomes from traditional scanning lithography and the proposed scanning lithography, and the captions should be more definitive to avoid confusion.

5.       Figure 9 requires more detailed explanations of the printing parameters, the process for measuring mechanical strength, and information on the instrumentation used to conduct these measurements.

By addressing these points, the paper can be strengthened and better contribute to the field of dynamic projection scanning lithography.

Extensive proof-reading and English editing are required. 

Reviewer 2 Report

In this article, it is mentioned that a new method is used to solve the problem of the stair effect, but it does not specifically explain and compare the details. What are the advantages?

The description of many details is not clear.  How to test the mechanical strength and discussion , in Figure 9 , there is no scale.  

Limitation of the method can be discussed.

Please consider providing a clearer description

Reviewer 3 Report

This manuscript developed a novel 3D printing method for larger, high-resolution structures called dynamic projection scanning lithography using a Digital Micromirror Device (DMD). The research is appropriately described. However, there are still a few issues that need to be resolved before publication. On page 5, Line 179 was left open without a punctuation mark, and Line 181 had a sentence that was incomplete. Additionally, it missed a space between "Dragon" and "measuring" on page 6 and line 203. Second, there were also gaps in the information regarding the resin used in Figures 7 and 8. Thirdly, there is a lack of information regarding the bending load test, including the equipment used, the load speed, and other testing parameters. Additionally, the detailed printing procedures used in Figure 9 with the name "step-by-step" were not accurately described in earlier sections in terms of the printing apparatus and the printing settings. And how do the loading direction and printing direction relate to one another? Fourthly, it would be preferable to include numerical references for the state of the art that was described in the introduction section, such as printing speed, printing resolution, overall build space, and cost. Although the SLA, LCD, and DLP processes have been reviewed and qualitatively compared, it is challenging to fully understand the benefits of the proposed approach. For instance, the author noted that DLP printing has a limited print capability for large-volume models. However, the Envison TEC Xtrem 8K machine, has a build volume of 450 x 371 x 399 mm, which is nearly the same as the proposed build space in this work of 400 x 400 x 200 mm. It would be preferable to include some further variables, like cost, to compare the effectiveness of the suggested strategy comparing with the state of the art. The research came to its final conclusion by saying that the suggested strategy may reduce the "staircase" effect on the printing outcomes. It would be preferable to compare the printing outcomes with other benchmark printing technologies, such as commercialized SLA and DLP printers. Although the oblique scanning technique was compared to conventional DMD scanning printing, it will be better to compare the printing outcomes with the other printers. 

Reviewer 4 Report

This paper introduces dynamic projection 3D printing technology utilizing Digital Micromirror Device. This technology can achieve large-size, high-precision structural printing in industrial fields.

The Reviewer believes that the following issues shall be addressed before accepting for publication:

Fig. 9 shows the bending test results of three different bending specimens. But no units are given in the figure, which needs to be supplemented. In addition, the dimensions and bending strength of the three specimens should be added.

The contents of lines 179-181 of the paper need to be revised.

Round 2

Reviewer 1 Report

The authors have responded to my comments and significantly revised the manuscript based on my suggestions. 

Extensive proof reading required.